# Novel Biomarkers for Early Detection of Hepatocellular Carcinoma

**DOI:** 10.3390/diagnostics14202278

**Published:** 2024-10-13

**Authors:** Abdelrahman M. Attia, Mohammad Saeid Rezaee-Zavareh, Soo Young Hwang, Naomy Kim, Hasmik Adetyan, Tamar Yalda, Pin-Jung Chen, Ekaterina K. Koltsova, Ju Dong Yang

**Affiliations:** 1Karsh Division of Gastroenterology and Hepatology, Cedars-Sinai Medical Center, Los Angeles, CA 90048, USA; abdelrahman.attia@cshs.org (A.M.A.); naomy.kim@cshs.org (N.K.); hasmik.adetyan@cshs.org (H.A.); tamar.yalda@gmail.com (T.Y.); 2Middle East Liver Diseases (MELD) Center, Tehran 1417935840, Iran; dr_rezaee@live.com; 3Department of Internal Medicine, University of Maryland Medical Center, Midtown Campus, Baltimore, MD 21201, USA; sooyoungsarah@gmail.com; 4Department of Hematology and Oncology, Cedars-Sinai Medical Center, Los Angeles, CA 90048, USA; pin-jung.chen@cshs.org; 5Cedars-Sinai Cancer, Smidt Heart Institute, Cedars-Sinai Medical Center, Los Angeles, CA 90048, USA; ekaterina.koltsova@cshs.org; 6Comprehensive Transplant Center, Cedars-Sinai Medical Center, Los Angeles, CA 90048, USA; 7Samuel Oschin Comprehensive Cancer Institute, Cedars-Sinai Medical Center, Los Angeles, CA 90048, USA

**Keywords:** liver neoplasms, hepatocellular carcinoma, early detection of cancer, biomarkers, diagnosis: screening, surveillance

## Abstract

Hepatocellular carcinoma (HCC) is a leading cause of cancer mortality globally. Most patients present with late diagnosis, leading to poor prognosis. This narrative review explores novel biomarkers for early HCC detection. We conducted a comprehensive literature review analyzing protein, circulating nucleic acid, metabolite, and quantitative proteomics-based biomarkers, evaluating the advantages and limitations of each approach. While established markers like alpha-fetoprotein (AFP), des-gamma-carboxy prothrombin, and AFP-L3 remain relevant, promising candidates include circulating tumor DNA, microRNAs, long noncoding RNAs, extracellular vesicle, and metabolomic biomarkers. Multi-biomarker panels like the GALAD score, Oncoguard, and Helio liver test show promise for improved diagnostic accuracy. Non-invasive approaches like urine and gut microbiome analysis are also emerging possibilities. Integrating these novel biomarkers with current screening protocols holds significant potential for earlier HCC detection and improved patient outcomes. Future research should explore multi-biomarker panels, omics technologies, and artificial intelligence to further enhance early HCC diagnosis and management.

## **1.** Introduction

Over 860,000 cases of liver cancer were diagnosed in 2022, making it the sixth most common cancer. Nearly 760,000 cancer deaths were due to liver cancer, which is the third leading cause of cancer death, accounting for 7.8% of all cancer deaths [1]. Between 75% and 85% of liver cancer cases are hepatocellular carcinoma (HCC) [2].

Most cases of HCC are associated with chronic liver disorders and cirrhosis, primarily resulting from hepatitis B virus (HBV) and C viruses (HCV) infections, alcohol-associated liver disease, and metabolic dysfunction-associated steatotic liver diseases (MASLD) or steatohepatitis (MASH) [3]. HBV and HCV account for about 55% and 21% of HCC cases, respectively [4]. Chronic inflammatory processes result in the development of liver fibrosis and cirrhosis. Ultimately, this may lead to the development of HCC. Regrettably, the majority of individuals with early-stage HCC do not exhibit any symptoms, resulting in the late diagnosis of HCC in the absence of a surveillance test [5].

Due to the frequent presentation of HCC in later stages, there is an urgent need to investigate newer strategies for detecting early-stage HCC [6]. The Early Diagnosis Research Network (EDRN) has introduced a five-phase approach to standardize research on novel cancer biomarkers, which have gained significant acceptance and implementation [7]. In 2021, the International Liver Cancer Association (ILCA) proposed customized EDRN biomarker stages specifically designed for HCC, which will be discussed in the subsequent sections [8].

In this review, we aim to discuss the use of various biomarkers for the diagnosis of HCC, including downregulated/upregulated proteins during HCC carcinogenesis, circulating nucleic acids or cells, metabolites, and the newly discovered biomarkers identified through quantitative proteomics. We delineate the constraints of the existing HCC screening strategy, examine the theoretical framework of precision medicine methods to surmount these obstacles, and provide an overview of upcoming advancements aimed at improving early HCC detection. Our particular emphasis is on novel biomarkers that are expected to impact the HCC screening program, ultimately increasing early-stage cancer detection and decreasing mortality.

## 2. Materials and Methods

In this narrative review, we searched multiple databases, including PubMed, Scopus, and Web of Science, to explore studies on novel biomarkers for the early detection of HCC. Keywords such as “Hepatocellular Carcinoma”, “Early Detection”, “Biomarkers”, “Protein Biomarkers”, “Circulating Nucleic Acids”, “Metabolomics”, and “Proteomics” were used to guide the search. The review included studies that met the following criteria: (1) published in peer-reviewed journals, (2) focused on novel biomarkers (e.g., protein, nucleic acid, metabolite, or quantitative proteomics-based biomarkers) for the early detection of HCC, and (3) available in English. Non-peer-reviewed articles and studies with insufficient methodological details were excluded. We categorized biomarkers into two groups, traditional and emerging biomarkers, and discussed the future of HCC biomarkers for early detection. For conflicting topics, we compared studies with differing results, and when meta-analyses were available, we included them in the discussion.

### 2.1. Challenges in Hepatocellular Carcinoma Screening

Detecting HCC at an early stage is crucial for enhancing patient outcomes. Surveillance programs have been put in place to identify individuals who pose a high risk for HCC. Effective HCC screening is still a difficult task, nevertheless, for a number of reasons. It might be difficult to accurately identify people who are at risk of developing HCC, often known as surveillance candidates. Moreover, the sensitivity of common surveillance tests, such as alpha-fetoprotein (AFP) and ultrasound, is limited, frequently resulting in missed diagnoses. Furthermore, patients with obesity or MASLD have poor diagnostic accuracy with ultrasonography. The changing etiology of HCC also makes it challenging to implement screening programs. The increasing prevalence of obesity and metabolic disorders worldwide has led to more cases of non-viral HCC, making traditional risk assessment models less reliable. These shifts in HCC require the development of new screening methods that are specific to different causes of the disease [3,9]. These drawbacks highlight the urgent need for better screening methods to enhance early HCC identification and improve patient outcomes [10,11,12].

A considerable proportion of HCC cases are discovered at advanced stages due to individuals with cirrhosis not undergoing surveillance, which contributes to the poor prognosis of HCC [11,12,13]. Performing abdominal ultrasonography every six months for HCC surveillance is linked to the early identification of tumors, receiving curative therapy, and improved overall survival [12]. However, fewer than 25% of individuals with cirrhosis who are at risk undergo HCC surveillance [14]. HCC surveillance is less commonly performed in racial and ethnic minority groups, including Black and Hispanic individuals, as well as those with lower socioeconomic status. This gap in surveillance contributes to the variations in HCC outcomes [15,16,17].

Efficient cancer screening necessitates a series of actions in a continuous process of care, which involves identifying individuals who are susceptible to cancer, healthcare providers advising and requesting screening, and patients complying with the timely completion of surveillance tests [18,19]. Surveillance failures can happen at any point across the range of factors relating to patients, clinicians, and the healthcare system [20]. Previous research examining obstacles to HCC monitoring has indicated that patient-related factors, such as logistical challenges (e.g., transportation), and clinician-related issues, such as limited knowledge and time restrictions in the clinic, are linked to the receipt of HCC surveillance [10,21,22].

Finally, despite its high sensitivity for detecting HCC overall, ultrasound with AFP fails to detect more than one-third of cirrhotic HCC cases at early stages [23]. Moreover, ultrasonography and AFP frequently produce false-positive or inconclusive outcomes, leading to possible physical, financial, and psychological damage [24].

The significant constraints of existing monitoring systems underscore the lack of a straightforward and effective approach to HCC surveillance. Therefore, there is a pressing need for the development of novel methods and strategies to improve the efficiency of HCC screening.

### 2.2. Precision in Hepatocellular Carcinoma Surveillance and Early Detection

To overcome the shortcomings of the existing HCC screening process and enhance its efficacy, it is necessary to improve the performance of risk stratification and early detection tests. Additionally, these tests should be logically integrated and arranged within a systematic HCC screening algorithm. To improve the efficiency of each test, the combination of clinical, molecular, and imaging factors has often been used for both risk stratification and early-stage disease detection. This approach aims to improve the effectiveness and practicality of early detection tests [25]. To prevent the failure of early detection and unnecessary harm from screening tests, it is important to use early detection tests based on the expected risk of HCC. This will help avoid underscreening high-risk patients and over-screening low-risk patients [26]. Precision medicine has the potential to improve both the early detection rate and precision of diagnosing HCC. Recent research has uncovered new information about the complex signaling regulatory network in HCC, leading to a better understanding of the disease and improving the accuracy of HCC detection [27].

To effectively execute screening, it is crucial to use an individualized, risk-based, and customized strategy. Utilizing methods like gene sequencing and omics technology to categorize HCC based on its molecular characteristics can contribute to the advancement of precision medicine. In addition, the rapid advancement of biocompatible nanomaterials, artificial intelligence, machine learning, data mining technologies, and interdisciplinary collaboration have greatly supported the field of radiomics and the development of new contrast agents. This has led to significant improvements in the accuracy of medical imaging for HCC diagnosis [27]. Utilizing next-generation sequencing (NGS) may help characterize the microenvironment of precancerous liver tissue. Nevertheless, the customized strategy should also provide a cost-efficient and accurate biomarker for patients who are at risk of HCC. The identification of biomarkers with high sensitivity and specificity is crucial for personalized surveillance strategies [28].

NGS is a potential approach for discovering the molecular basis of HCC, allowing for the detection of new biomarkers. By customizing surveillance strategies according to specific patient features and genetic profiles, we can improve the early diagnosis of HCC and enhance patient outcomes. Successful precision medicine in HCC surveillance will require overcoming hurdles associated with refined risk stratification and implementation [29].

### 2.3. Biomarker Development for Early Detection of Hepatocellular Carcinoma

Screening is essential for the early identification and successful management of HCC. Clinical symptoms, imaging, and serum protein biomarkers are used in detecting HCC [30]. The serum protein biomarkers often used for HCC include AFP, des-γ-carboxyprothrombin (DCP) [31], and glypican-3 (GPC-3) [32]. The presence of heterogeneity in HCC greatly hampers the ability to diagnose and treat individuals with this condition [33,34], resulting in poor diagnostic and therapeutic efficiency. With the emergence of NGS and the advancements in precision medicine, omics data, such as genomics, epigenomics, transcriptomics, proteomics, and metabolomics, can detect biological heterogeneity, facilitating the discovery of novel HCC biomarkers, as seen in Figure 1.

This figure illustrates the integration of various omics data in the development of early detection biomarkers for HCC. The components include (1) ctDNA, miRNA, circRNA, and lncRNA. These represent different types of nucleic acids found in the bloodstream, which can be sequenced and analyzed to identify tumor-specific mutations, expression patterns, or other biomarkers indicative of HCC. (2) Tumor Protein Markers: Proteins produced by tumor cells can serve as biomarkers detectable in blood samples. (3) Extracellular Vesicles (EVs), Red Blood Cells (RBCs), Circulating Tumor Cells (CTCs), and Lymphocytes: These circulating components can carry tumor-specific markers, which can be analyzed to detect the presence of HCC. (4) HCC Tissue: Direct analysis of HCC tissue provides critical insights into the molecular and genetic characteristics of the tumor, contributing to biomarker discovery. (5) Intestinal Flora: The composition and interactions of the gut microbiome with the host can influence liver disease and potentially serve as indirect biomarkers for HCC.

EDRN established five widely accepted stages for developing early cancer detection biomarkers [7] and, in 2021, ILCA customized these stages specifically for HCC [8]. Phase 1 is a preliminary investigation conducted in preclinical environments to identify possible biomarkers. Typically, this procedure involves comparing tumor tissue with non-tumor tissue using equipment that may identify protein or gene expression. The goal of a Phase 1 gene expression or proteomics investigation is to detect genes or proteins that show higher or lower levels of expression in tumor tissue compared to control tissue. However, the harvesting of organ tissue is typically too intrusive to be used for clinical screening purposes. Therefore, the next phase involves developing a clinical test that relies on measuring either the quantities of proteins produced by the identified genes in the serum or the level of antibodies against those proteins in the serum. Phase 2 involves developing a clinical test by comparing samples from individuals with malignancies to those without. The procedure is conducted in a clinical environment, often through a case-control study, following the diagnosis of cancer in individuals. The purpose of this phase is to calculate the true positive rate (TPR) and false positive rate (FPR) and to assess the area under the receiver operating characteristic curve (AUROC) of the clinical biomarker assay, evaluating its capacity to differentiate individuals with cancer from those without. Phase 3 is a retrospective longitudinal study designed to assess the early detection capability of biomarkers and to establish the criteria for a positive screening test in preparation for Phase 4. This study contrasts individuals who have developed cancer with those who have not, utilizing clinical assays conducted at different time points. Phase 4 is a study designed to assess the effectiveness of biomarkers as a screening tool in a specific population. Phase 5 is a meticulously regulated investigation, preferably a randomized controlled trial, that includes therapeutic intervention if necessary. Its purpose is to assess the mortality advantage of biomarker screening using a positive screening test. A summary is shown in Figure 2.

### 2.4. Classification of Biomarkers

The ideal biomarker for routine clinical analysis should possess several key properties. It should be sensitive and specific and not require extensive operator experience. Additionally, it should be inexpensive, highly reproducible, and capable of producing rapid results [35]. Finally, the biomarker should also correlate with tumor stages and be easily obtainable without the need for pretreatment of the available samples, such as blood or urine, as shown in Figure 3. The categorization of HCC markers is detailed in Table 1.

### 2.5. Traditional Serum Protein Biomarkers

#### 2.5.1. Alpha-Fetoprotein

The serum AFP is used as a biomarker for screening and monitoring of therapeutic response for HCC. AFP is a glycoprotein detectable in the blood of fetuses and is the first acknowledged oncofetal biomarker. Its usefulness in liver cancer diagnostics dates back to its initial identification in mouse hepatoma and subsequent confirmation in the serum of patients with HCC [36,37]. As far back as the 1960s, AFP was identified as a biomarker not only for HCC but also for distinguishing between primary and metastatic liver tumors [38]. Following that, sequential biomarker investigations have examined the utility of AFP for the identification and screening of HCC [37]. The AUROC for AFP was 0.77, with a sensitivity and specificity of 62% and 87%, respectively [39]. The blood AFP level in healthy persons is less than 5 ng/mL. Conversely, a high concentration of AFP in the blood is often linked to HCC or other inflammatory liver conditions. However, a significant percentage of patients with HCC may have normal AFP levels, even in advanced stages of the disease, limiting its role as a standalone test for HCC surveillance [40].

#### 2.5.2. Des-γ-Carboxy Prothrombin

DCP is a variant form of prothrombin that is deficient in γ-carboxy residues, resulting in impaired clotting ability. This protein is activated in response to a deficiency or inhibitor of vitamin K, known as PIVKA II. Malignant hepatocytes have a deficiency in carboxylating glutamic acid to produce γ-carboxy glutamic acid. DCP, also known as aberrant prothrombin, is produced due to this malfunction [41]. When comparing AFP and DCP levels, it was shown that DCP levels had higher sensitivity and specificity in differentiating between HCC and chronic nonmalignant hepatic disorders. The sensitivity of HCC detection was enhanced by combining the use of DCP and AFP [42,43]. In a study that included 689 patients with cirrhosis and/or chronic hepatitis B, as well as 42 cases of HCC, matched analysis showed that the AUROC for DCP was 0.7 at HCC diagnosis [44].

#### 2.5.3. Alpha-Fetoprotein-L3

AFP-L3 is a subtype of AFP that is found in cancerous liver cells and is considered to be specific to HCC [45]. By measuring the proportion of AFP-L3 to total AFP (AFP-L3%), it is possible to diagnose HCC at an early stage. Previous studies have shown that AFP-L3% has a diagnostic sensitivity for HCC ranging from 75.0% to 96.9%, with a specificity of from 90.0% to 92.0% [46,47]. In a study involving 689 patients with cirrhosis or chronic HBV, 42 who progressed to HCC were matched with 168 controls. The AUROCs for AFP, AFP-L3, and DCP at diagnosis were 0.77, 0.73, and 0.71, respectively. Combining AFP and AFP-L3 improved the AUROC to 0.83, but adding DCP only slightly increased it to 0.86. The optimal cutoff values for AFP (5 ng/mL) and AFP-L3 (4%) yielded a sensitivity of 79% and specificity of 87% for HCC detection [39].

## 3. Emerging Biomarkers

### 3.1. Nucleic Acid Biomarkers

#### 3.1.1. Circulating Tumor DNA (ctDNA)

In genomics, ctDNA is a valuable diagnostic tool for cancer. It is minimally invasive, requiring only a small blood sample, and can reveal genetic and epigenetic changes associated with cancer and its spread. ctDNA provides an easy and accurate method for ongoing tumor genome surveillance [48]. In the field of epigenomics, the examination of methylation patterns in ctDNA shows potential for the detection of HCC [49]. Building on this potential, several blood test panels were developed for early liver cancer detection using ctDNA analysis. HelioLiver Test combines circulating cell-free DNA methylation markers with patient demographic data and established HCC tumor markers. In a 2022 study, the HelioLiver Test was assessed in 247 participants (122 with HCC and 125 with chronic liver disease). The test demonstrated an AUROC of 0.944, outperforming AFP (0.851; *p* < 0.0001) and GALAD (0.899; *p* < 0.0001). It achieved 85% sensitivity (95% CI: 78–90%) for any-stage HCC, 76% sensitivity (95% CI: 60–87%) for early-stage HCC, and 91% specificity (95% CI: 85–95%), indicating its superior performance and potential for effective HCC surveillance [50]. Similarly, the Oncoguard Liver test, which combines methylation biomarkers such as HOXA1, TSPYL5, and B3GALT6 with sex and AFP, demonstrated a sensitivity of 72% (95% CI: 61–80%) and specificity of 88% (95 CI: 84–91%) for detecting early-stage HCC in the algorithm development study. The validation study showed 82% (95% CI: 72–89%) sensitivity and 87% (95% CI: 82–91%) specificity. Like the HelioLiver Test, this test outperforms AFP and GALAD for early-stage detection but has lower specificity compared to these tests [51].

These commercially available tests highlight the growing clinical utility of ctDNA as a minimally invasive approach for the early detection and surveillance of liver cancer.

#### 3.1.2. MicroRNAs (miRNAs), Long Noncoding RNA (lncRNAs), and Circular RNA (circRNAs)

Transcriptomics has shown distinct expression patterns of circulating miRNAs in different tumors, including HCC [52,53,54]. The diagnostic value of three different types of RNA for HCC diagnosis was assessed in several studies. Several studies have demonstrated the diagnostic potential of specific miRNAs for HCC. For example, miR-122, miR-21, and miR-221 have been extensively studied, with findings indicating their elevated levels in HCC patients compared to healthy controls [55,56]. A meta-analysis reported that the diagnostic accuracy of miRNAs for HCC was comparable to traditional biomarkers, with sensitivity, specificity, and AUROC values exceeding 80% [57]. A network meta-analysis determined that circular RNA (circRNAs) is the most effective, with lncRNAs and miRNAs ranked second and third, respectively [58]. Several meta-analyses have assessed the diagnostic value of circRNAs in HCC diagnosis, reporting specificity, sensitivity, and AUROC values of at least 74%, 72%, and 0.815, respectively [59,60,61].

lncRNAs typically consist of about 200 nucleotides and play roles in maintaining the integrity of RNA and in binding to proteins and DNA. Several lncRNAs circulating in the blood have shown diagnostic potential for HCC. Li et al. [62] demonstrated that the combination of circulating lncRNAs of highly upregulated liver cancer (HULC) and Linc00152 resulted in an AUROC of 0.87 for HCC diagnosis. Furthermore, when these lncRNAs were combined with AFP, the AUROC increased to 0.89. Another study used a mix of three plasma lncRNAs—LINC00152, RP11-160H22.5, and XLOC014172—and found AUROC values of 0.985 and 0.986 for distinguishing between healthy individuals or those with chronic hepatitis and those with HCC [63]. Furthermore, other lncRNAs, such as p34822, have also been shown to possess diagnostic significance [64].

A 2021 meta-analysis involving approximately 5000 patients and 4600 controls evaluated the diagnostic value of lncRNAs for HCC, reporting pooled sensitivity, specificity, and AUROC values of 0.85 (95% CI: 0.82–0.88), 0.76 (95% CI: 0.73–0.80), and 0.88 (95% CI: 0.85–0.91), respectively [65].

Another meta-analysis, published in 2024, focused on the serum levels of three lncRNAs, HULC, urothelial carcinoma-associated 1 (UCA1), and homeobox transcript antisense intergenic RNA (HOTAIR), and found that these lncRNAs, when evaluated together, offered significantly better sensitivity and specificity for diagnosing HCC than traditional biomarkers or other ncRNAs. These results underscore the potential of these lncRNAs to improve early detection and support personalized treatment strategies for HCC [66].

### 3.2. Metabolomic Biomarkers

Metabolomics, a branch of the “omics” technique, enables the reliable identification, quantification, and characterization of small metabolites with molecular masses below 2 kDa. HCC is characterized by a variety of cellular metabolic changes. These modifications result in the formation of identifiable substances in the bloodstream, including bile, phospholipids, peptides, sphingolipids, reactive oxygen species, amino acids, long-chain carnitines, and modified nucleosides like 1-methyladenosine (M1A), among others [67]. Hepatocellular metabolic changes are of utmost importance in the development of cancer [35]. For example, in tumors, the glycolysis process is redirected to produce nucleotides that are needed for the pentose-phosphate pathway instead of generating energy in the form of ATP. Nevertheless, comprehending the metabolomics of tumors presents a promising strategy for identifying possible biomarkers that might aid in the early detection of HCC [68,69,70]. Metabolomics analysis of 612 blood samples from 203 MRI-monitored cirrhotic patients, 37 of whom developed HCC, identified six key markers that effectively distinguished patients with HCC from those without it within a year. The markers (AFP, 6-bromotryptophan, N-acetylglycine, salicyluric glucuronide, testosterone sulfate, and age) achieved an AUROC of 0.88 (95% CI: 0.83–0.93). This study highlighted N-acetylglycine, certain amino acids, bile acids, and choline-related metabolites as potential HCC risk biomarkers [71].

### 3.3. Extracellular Vesicles (EV)

The presence of a lipid layer in EV prevents the degradation of mRNA, microRNA, and lncRNA by RNases. Multiple studies have shown that serum EV-containing RNAs can serve as biomarkers for screening HCC [53,72]. Xu et al. compared serum levels of the EV markers ENSG00000258332.1 and LINC00635 among HCC patients, chronic HBV-infected patients, and healthy controls. They found that these markers were elevated in HCC patients and decreased after surgery. Combining ENSG00000258332.1, LINC00635, and AFP achieved a high diagnostic accuracy with an AUROC of 0.894, sensitivity of 83.6%, and specificity of 87.7%. ENSG00000258332.1 and LINC00635 individually had AUROCs of 0.719 and 0.750, respectively, for distinguishing HCC from chronic HBV [73]. Wang et al. utilized deep sequencing to analyze serum EV miRNA profiles and compared them among groups with HCC, liver cirrhosis, and healthy controls. They found that EV miRNA-148a had an AUROC of 0.891 (95% CI: 0.809–0.947) for differentiating HCC from liver cirrhosis, while AFP had an AUROC of 0.712 (95% CI: 0.607–0.803). The combined use of EV miR-122, miR-148a, and AFP improved the AUROC to 0.931 (95% CI: 0.857–0.973), making it effective for distinguishing early HCC from liver cirrhosis. Among these markers, miR-122 was particularly effective in differentiating HCC from healthy controls, with an AUROC of 0.990 (95% CI: 0.945–1.000) [74].

In addition, serum EV heterogeneous nuclear ribonucleoprotein H1, LINC00161, and miRNA224 were shown to effectively differentiate between patients with HCC and healthy individuals, with AUROC values of 0.865, 0.794, and 0.91, respectively [75,76,77].

LINC00853, present in EVs, was found in 97% of people with early HCC who had negative AFP tests and in 67% of those with positive AFP tests [78]. The expression levels of Long Intergenic Non-Protein Coding RNA 853 (LINC00853) and miR-10b-5p were increased in HCC tissues and extracellular EVs [79]. The corresponding AUROC values were 0.96 and 0.94 for detecting early-stage HCC (single lesion: <2 cm), respectively [78,80].

There are also other encouraging studies indicating that EV-based biomarkers could play a role in the early diagnosis of HCC. RNA sequencing of EVs revealed the presence of three distinct clusters of short RNA molecules. These mostly unannotated short cluster RNAs, characterized by a 20-nucleotide sequence, were significantly elevated in HCC patients. In the phase 2 biomarker case-control study, performed on 105 patients with early-stage HCC (BCLC stage 0/A) and 85 patients with chronic liver disease [81], these clusters demonstrated high sensitivity, specificity, and AUROC values of 86%, 91%, and 0.87, respectively. In another study, the HCC EV surface protein assay (combining covalent chemistry for EV purification with duplex real-time immune PCR for quantification) analyzed eight HCC EV subpopulations in 400-microliter plasma samples. By measuring these subpopulations (epithelial cellular adhesion molecule+ CD63+ HCC EVs, CD147+ CD63+ HCC EVs, and Glypican 3 Protein+ CD63+ HCC EVs), a logistic regression model developed an HCC EV ECG score to distinguish early-stage HCC from cirrhosis. The phase 2 biomarker study found that this ECG score achieved an AUROC of 0.95 (95% CI: 0.90–0.9) in the training cohort and 0.93 (95% CI: 0.87–0.99) in the validation cohort, highlighting its effectiveness for early HCC detection [82]. In a 2020 study, Sun et al. developed a unique EV mRNA panel utilizing a combination of microfluidics and reverse-transcription droplet digital PCR, and it showed a sensitivity of 94%, specificity of 89%, and an AUROC of 0.93 (95%CI: 0.86–1.00) for early detection of HCC among at-risk cirrhotic patients. This was observed in a group of 36 patients with early-stage (BCLC stage 0/A) HCC and 26 cirrhosis controls [83]. Furthermore, a meta-analysis of 18 studies, published in 2023, analyzed the effectiveness of serum-derived EV for diagnosing HCC. The analysis looked at four types of biomarkers: EV miRNAs, EV RNAs, AFP, and a combination of EV RNAs with AFP. EV miRNAs had the highest sensitivity (0.86) and the lowest negative likelihood ratio (0.17). The combination of EV RNAs and AFP had the highest specificity (0.89), positive likelihood ratio (7.55), diagnostic odds ratio (35.96), and AUROC (0.93). This study showed that EV biomarkers perform better than AFP alone, and combining them yields the best diagnostic results [84]. Overall, while these studies highlight the potential of EVs as early detection biomarkers for HCC, larger studies are still needed.

### 3.4. Biomarker Panels

Although several attempts have been made to find optimal biomarkers, there is currently no single biomarker with high sensitivity for HCC. When used with other clinical indicators, the identification of some biomarkers exhibits greater sensitivity and specificity compared to using a single biomarker alone. Thus, more recent biomarkers and models, such as Lens culinaris agglutinin-reactive fraction of AFP, AFP-L3, DCP, or PIVKA-II, and the GALAD score, are being used for early-stage HCC detection [85].

The GALAD score is a biomarker panel-based model that was developed to assess the risk of HCC in high-risk patients. It takes into account the individual’s gender, age, and levels of AFP, AFP-L3, and DCP [86]. The GALAD algorithm has shown notable accuracy in HCC detection, notably in individuals with cirrhosis, irrespective of the etiology of their liver illness [85].

GALAD score was validated in the US in a study published in 2019, where the authors found that the AUROC of the GALAD score for detecting HCC was 0.88 (95% CI: of 0.85–0.91) in the EDRN multicenter prospective cohort. Furthermore, the authors also introduced the GALADUS score, which combines the GALAD with ultrasound to improve its effectiveness. This new score achieved an AUROC of 0.98 (95%CI: 0.96–0.99) with a cut-off of −0.18, showing a sensitivity of 95% and a specificity of 91% [87].

The GALAD score has also been evaluated in studies conducted in various other countries. In a multicenter study that included over 2000 patient samples from six Latin American and two European countries, researchers assessed the effectiveness of the GALAD score in distinguishing HCC from cirrhosis. In Latin American patients, the GALAD score achieved an AUROC of 0.76, with a sensitivity of 70% and specificity of 83%, while, in a European cohort, it showed an AUROC of 0.69, with 66% sensitivity and 72% specificity. The study also explored an optimized version of the score, revealing that AFP-L3 had minimal impact on early-stage HCC detection. This led to the creation of the ASAP score, which excludes AFP-L3. The ASAP score demonstrated significant potential for early-stage HCC detection and could identify cirrhotic patients at high risk for advanced HCC up to 15 months before diagnosis. Additionally, it successfully differentiated HCC from hemangiomas, with 100% specificity at 71% sensitivity [88].

Another notable score is the HCC Early Detection Screening (HES) score, which incorporates AFP, age, alanine aminotransferase, and platelet count. This score has shown improved early detection of HCC compared to AFP alone. In an effort to find a better alternative to GALAD and ASAP, researchers developed an updated version of HSE called HES 2.0. This new version includes AFP-L3 and DCP to improve accuracy and was evaluated in a prospective cohort study of 2331 patients with cirrhosis, including 125 who developed HCC (71% of which were at an early stage). The study demonstrated that HES 2.0 outperformed both GALAD and ASAP scores in terms of sensitivity or TPR for early HCC. Specifically, HES 2.0 had higher sensitivity/TPR compared to GALAD overall (+6.7%) at 12 months (+6.3%) and at 24 months (+14.6%), though it was similar to GALAD at 6 months (+0.0%). Additionally, HES 2.0 showed higher sensitivity/TPR compared to ASAP at all evaluated time points (+13.4% to +18.0%) [89].

In summary, combining multiple biomarkers, as seen in the GALAD and HES 2.0 scores, offers better early-stage HCC detection than using a single biomarker, with HES 2.0 showing the best performance.

### 3.5. Urine-Based Biomarkers

The advancement of highly sensitive omics profiling technology has made it possible to analyze different cancer-related molecular information in bodily fluid samples, such as blood and urine [79]. Urine testing is a noninvasive technique that has been extensively investigated as a biomarker in human diseases [90]. The samples may be simply collected, transported, and kept. Urine, being a bio-product of blood filtration, collects unusual waste substances from the body’s circulation. This includes early signs of cancer development, which may be more plentiful and easier to detect in urine compared to blood [91]. The urine samples are more resistant to environmental fluctuations and less susceptible to disruption or contamination throughout the inspection processes [91]. Although the overall concentration of proteins, nucleic acids, and other compounds is reduced in urine compared to blood, this reduction in background noise can make it easier to accurately detect biomarkers owing to the decreased level of background noise. Recently, there has been a growing interest in urinary biomarkers for HCC detection. Certain biomarkers discovered via these studies have shown significant potential not only in the areas of HCC diagnosis but also in therapy, monitoring, and prognosis [92].

A Phase 1/2 multicenter trial was conducted to assess a urine ctDNA panel consisting of a TP53 mutation and two methylation markers, mRASSF1A and mGSTP1. The panel was examined in a total of 279 patients with chronic hepatitis B, 144 patients with cirrhosis, and 186 patients with HCC [93]. The ctDNA panel did not perform better than AFP, with an AUROC of 0.74 and 0.85, respectively. Nevertheless, the use of a two-step approach, including the first application of AFP followed by the utilization of the ctDNA panel in patients with AFP levels below 20 ng/mL, resulted in a significant enhancement of the AUROC to 0.91. The sensitivities for identifying BCLC stage 0 and stage A HCC using this two-step technique were 92% and 77%, respectively, with a fixed specificity of 90%. Lin et al. investigated the use of cell-free DNA to identify methylation markers associated with HCC. They analyzed samples from 31 non-HCC and 30 HCC patients, identifying 29 genes with differential methylation. Methylation-specific qPCR validated significant changes in four genes (GRASP, HOXA9, BMP4, and ECE1) between HCC and non-HCC patients. In a separate cohort of 87 non-HCC and 78 HCC patients, a 6-marker panel, including GSTP1 and RASSF1A, achieved an AUROC of 0.908 for HCC detection, surpassing the AUROC of 0.841 for AFP alone. Furthermore, the 4-marker panel also showed comparable results, with 80% sensitivity versus AFP’s sensitivity of 29.5% and a specificity of 85% [94].

In a study focused on improving the early detection of HBV-related HCC, miR-93-5p emerged as a promising biomarker, identified not only in HCC tissue and plasma but also in the urine of patients with early HBV-related HCC. Urine miR-93-5p, with an AUROC of 0.901, demonstrated 87.5% sensitivity and 97.4% specificity for early HCC detection. Importantly, there was no significant difference between plasma and urine miR-93-5p in detecting early HBV-related HCC. However, the use of healthy controls may have led to an overestimation of detection power, and the small number of samples from cases with HBV-related HCC, along with the restriction to a Chinese Han population, were the main limitations. Therefore, further multinational studies are needed to confirm these findings [82].

Surface-enhanced Raman spectroscopy (SERS) is a powerful analytical technique that amplifies the Raman scattering signals of molecules adsorbed on nanostructured metallic surfaces, such as gold or silver. This amplification enables the detection of molecular compositions at very low concentrations, making SERS highly sensitive for both chemical and biological analyses. In a study published in 2022, SERS spectra were recorded from the urine of 49 liver cirrhosis patients, 55 HCC patients, and 50 healthy volunteers using a Raman spectrometer. The analysis, combined with a support vector machine (SVM) algorithm, revealed distinct differences in specific biomolecules and metabolic profiles associated with liver cirrhosis and HCC. The urine SERS method demonstrated strong performance in distinguishing HCC, achieving a sensitivity of 85.5%, specificity of 84.0%, and accuracy of 84.8%. Compared to serum AFP, the urine SERS method exhibited greater diagnostic sensitivity for HCC, reaching up to 90% [95]. In addition, a 2022 study explored using urine fluorescence spectroscopy with machine learning to differentiate between HCC, liver cirrhosis, and healthy controls. Urine samples from 62 HCC patients, 65 people with liver cirrhosis, and 60 healthy individuals were analyzed using a fluorescent scan multimode reader with a 405 nm excitation wavelength. The study found clear differences in certain metabolites, particularly abnormal levels of porphyrin derivatives and bilirubin in those with HCC and liver cirrhosis. Using the SVM algorithm on these data, the approach reached an overall diagnostic accuracy of 83.42%. It showed a sensitivity of 93.6% for detecting HCC and 73.9% for liver cirrhosis, with specificities of 88.0% and 89.3% for each condition, respectively [96].

Due to its practical advantage in sample accessibility, urine is expected to remain a viable source of molecular information for early diagnosis of HCC once its accuracy is validated in larger prospective studies.

### 3.6. Gut Microbiome

The microbiome plays a crucial role in maintaining health and influences diseases by regulating vital inflammatory, metabolism, and immune responses. HCC risk could be modulated by the gut microbiome via interactions between the gut and liver. This has been shown in both animal models and human investigations. Therefore, biomarkers derived from gut microbiota may show great potential as non-invasive methods for the early detection of HCC [97].

Several studies have sought to determine the composition of the gut microbiome in HCC cases with different etiologies, including in viral and non-viral groups [98], HBV [99], HCV [100], MASLD [101], and in combined MASLD and HCV cases [102]. In cases of HCC associated with MASLD-related cirrhosis, patients exhibited higher levels of fecal calprotectin and plasma levels of IL-8, IL-13, CCL3, CCL4, and CCL5 compared to those with MASLD-induced cirrhosis without HCC. These HCC patients also had increased levels of Bacteroides, Enterococcus, and Ruminococcaceae, while Bifidobacterium levels were reduced. [103] Ex vivo studies indicated that bacterial components of microbiota from the MASLD-HCC triggered a T cell immunosuppressive response, characterized by an increase in regulatory T cells and a decrease in CD8+ T cell activity. This highlights the gut microbiota’s role in modulating the peripheral immune response in MASLD-HCC [101]. Using a mouse model lacking the inflammasome sensor molecule NLRP6, Schneider et al. demonstrated that dysbiotic microbiota negatively impacted immune surveillance against tumors by inducing a Toll-like receptor 4 (TLR4) dependent expansion of hepatic monocytic myeloid-derived suppressor cells (mMDSCs) and reducing T-cell abundance. This inflammatory microenvironment accelerated liver disease progression towards HCC, but the effects are reversible with antibiotic treatment or by reintroducing the beneficial bacterium Akkermansia muciniphila [104].

The gut microbiota is reported to increase from cirrhotic cases to early HCC cases [105,106]. Ren and colleagues investigated the gut microbiome in HCC patients and its potential as a non-invasive diagnostic tool. Analyzing 486 fecal samples from various regions in China, they focused on 75 early HCC patients, 40 cirrhotic patients, and 75 healthy controls. The study revealed increased microbial diversity from cirrhosis to early HCC, with notable elevations in Actinobacteria and specific genera like Gemmiger and Parabacteroides in early HCC. Butyrate-producing bacteria were reduced while lipopolysaccharide-producing bacteria increased. A model with 30 microbial markers achieved an AUROC of 80.6% for distinguishing early HCC from non-HCC, showing strong diagnostic potential across regions [105]. Similarly, a more recent study analyzed fecal samples from patients with hepatitis, cirrhosis, HCC, and healthy controls. They observed significant differences in gut microbial diversity, with notable increases from cirrhosis to HCC, particularly in cirrhosis-induced HCC cases. Thirteen bacterial genera were linked to tumor size, and three genera (Enterococcus, Limnobacter, and Phyllobacterium) were identified as potential biomarkers for HCC. The study found that dysbiosis was more prevalent in liver cirrhosis-induced HCC, marked by decreased butyrate-producing bacteria and increased lipopolysaccharide-producing bacteria [106]. Another study analyzed gut microbiomes of 30 patients with HCC-associated cirrhosis (HCC–cirrhosis), 38 with cirrhosis without HCC, and 27 healthy controls using 16S rRNA sequencing. Both cirrhosis groups had lower bacterial richness compared to healthy controls, but HCC–cirrhosis patients exhibited a distinct microbial profile with higher Clostridium and CF231 and lower Alphaproteobacteria abundance. The HCC–cirrhosis group was accurately classified from healthy controls with 82% accuracy and an AUC of 0.9 using a random forest classifier. Key discriminatory features included Veillonella dispar, Faecalibacterium prausnitzii, and Ruminococcus gnavus [102].

Focusing on chronic HBV cases, one study investigated the gut microbiome to identify biomarkers for diagnosing cirrhosis and HCC in a Chinese population. Researchers identified 14 cirrhosis-associated and 10 HCC-associated bacterial genera that significantly differed from healthy controls. These findings were used to develop random forest models that accurately distinguished cirrhosis and HCC from healthy individuals, with AUROC values of 0.824 and 0.902, respectively. The models’ accuracy improved further when clinical factors were included, highlighting their potential for early diagnosis of liver cirrhosis and HCC in chronic HBV cases [107]. The bacterial composition also proves useful in distinguishing between healthy individuals and different types of liver cancer. A classification model using eight key bacterial genera achieved high diagnostic accuracy (AUC = 0.989 for healthy controls, 0.967 for HCC, and 0.920 for cholangiocarcinoma). Furthermore, increased gram-negative bacteria and higher inflammatory markers were noted in cholangiocarcinoma compared to HCC [108].

In summary, the gut microbiome exhibits distinct alterations associated with liver cancer progression and type, demonstrating its potential as a non-invasive diagnostic tool. Elevated microbial diversity and specific bacterial profiles, along with associated inflammatory markers, highlight the gut microbiome’s role in early detection and differentiation of liver diseases, offering promising avenues for future diagnostic and therapeutic strategies, summarized in Table 2.

## 4. The Future of Hepatocellular Carcinoma Biomarker for Early Detection

With advances in multiomic approaches, new biomarkers will become accessible for the early detection of HCC. Obstacles persist in biomarker investigations, such as incomplete cohort data, biased sample collection, and small sample sizes for both initial discovery and subsequent validation studies. Due to the characteristics of HCC, the intricate nature of the illness, and the diverse range of risk factors, it is challenging to pinpoint and use a singular and universally applicable biomarker for the detection of early-stage HCC. Advancements in artificial intelligence and machine and deep learning methods, such as the integration of biomarkers with multiple features, have the potential to significantly improve the accuracy of prediction and diagnosis in the near future. The efficacy of this novel technique relies on the accessibility and availability of various types of information, including human genetic data and original laboratory and clinical data [115,116,117].

## 5. Conclusions and Perspectives

The advancement of omics technology has led to the discovery of several new biomarkers for HCC. Nevertheless, most of these novel biomarker candidates are based on preliminary research. Validating these biomarkers will require rigorous phase 3 or phase 4 studies with large sample sizes. Importantly, the concurrent use of multiple biomarkers, in conjunction with omics-based technologies and artificial intelligence, has the potential to enhance early identification and ultimately improve the dismal prognosis of HCC.

## Figures and Tables

**Figure 1 diagnostics-14-02278-f001:**
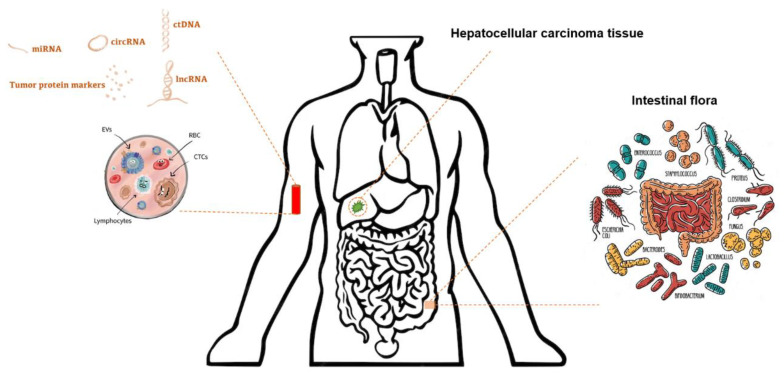
Development of early detection biomarkers for hepatocellular carcinoma using “omics” data.

**Figure 2 diagnostics-14-02278-f002:**
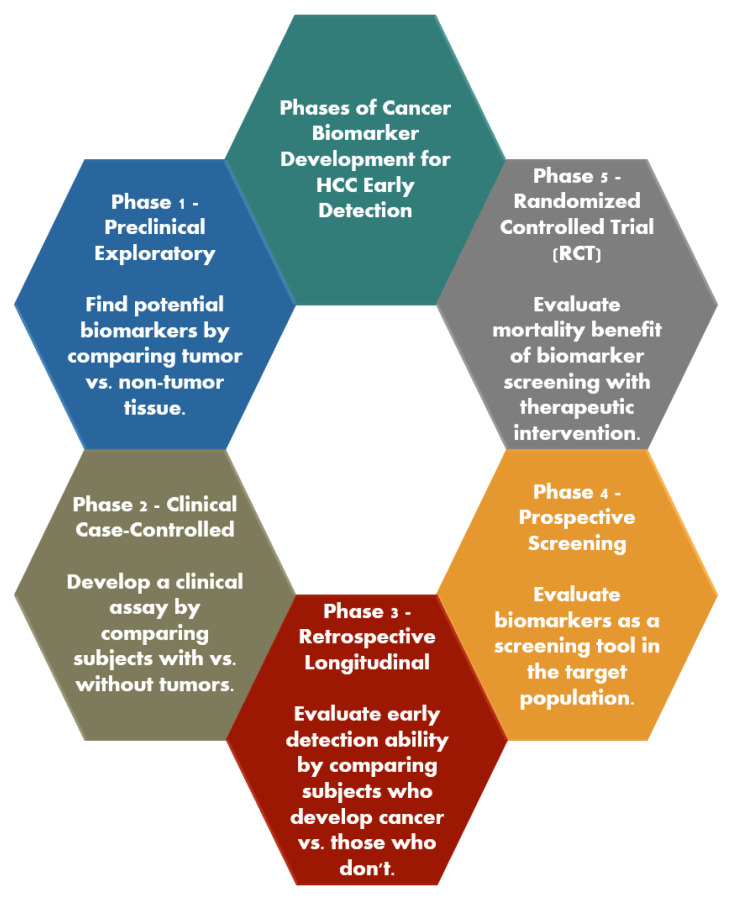
Phases of Cancer Biomarker Development for Early Detection of Hepatocellular Carcinoma: This figure summarizes the sequential phases involved in the development of cancer biomarkers specifically for the early detection of HCC: (1) Phase 1—Preclinical Exploratory: This initial phase involves identifying potential biomarkers by comparing tumor tissues with non-tumor tissues. The focus is on discovering gene expressions or protein markers that are differentially expressed in tumor tissues. (2) Phase 2—Clinical Case-Controlled: In this phase, a clinical assay is developed by comparing samples from individuals with and without HCC. The aim is to evaluate the true positive and false positive rates and the assay’s ability to differentiate between those with and without the disease. (3) Phase 3—Retrospective Longitudinal: This phase evaluates the early detection ability of the biomarkers. It involves a retrospective study comparing individuals who developed HCC with those who did not, using previously collected clinical samples. (4) Phase 4—Prospective Screening: The biomarkers identified in earlier phases are tested as screening tools in a target population. This phase evaluates their effectiveness in detecting HCC in a broader, real-world setting. (5) Phase 5—Randomized Controlled Trial: The final phase involves a controlled trial to assess the mortality benefit of using the biomarkers for screening, potentially coupled with therapeutic interventions. This phase determines the clinical utility of the biomarkers in improving patient outcomes.

**Figure 3 diagnostics-14-02278-f003:**
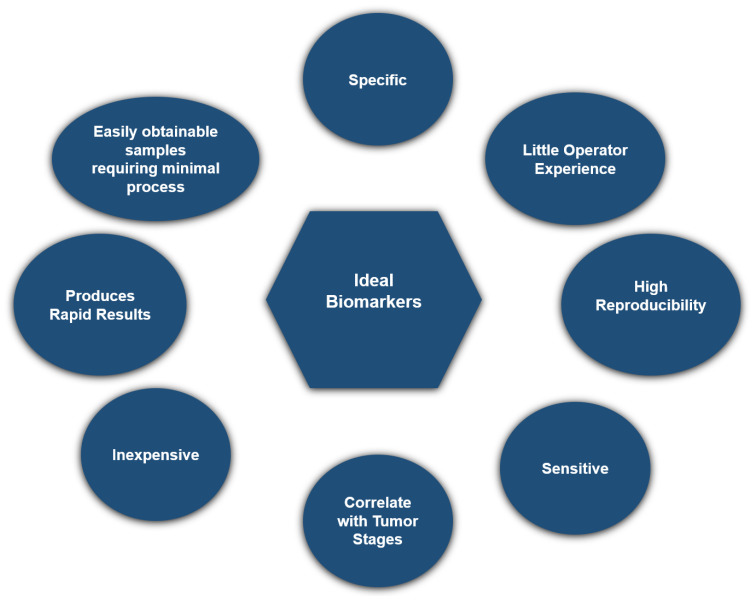
Characteristics of Ideal Biomarkers: This figure highlights the key attributes of an ideal biomarker for clinical use, including specificity, sensitivity, reproducibility, and cost-effectiveness. The biomarker should be easily obtainable with minimal processing, correlate with tumor stages, and provide rapid results without requiring extensive operator expertise.

**Table 1 diagnostics-14-02278-t001:** Classification of biomarkers for hepatocellular carcinoma diagnosis.

Category	Biomarker Examples	Advantages	Limitations
Proteins	AFP, AFP-L3, DCP, Glypican-3, osteopontin, GALAD score,	Widely available assays, non-invasive or minimally invasive collection, relatively simple detection methods	Low sensitivity and specificity for early-stage HCC—Elevated levels in non-HCC conditions
Emerging Biomarkers	Nucleic Acid Biomarkers: miRNAs and lncRNAs, ctDNA	Potentially high sensitivity and specificity, potential for personalized medicine	Extracting and examining certain nucleic acids from blood may be technically demanding and need specialized equipment.
	Exosomes, EV-associated biomarkers, Integrated Omics	Non-invasive, tumor specificity, stability, targeted therapeutics	These methodologies are currently in the process of being developed, and more investigation is required to verify their efficacy in clinical environments.
	Metabolites: amino acids, bile acids	Non-invasive approach, early detection potential, potential for multi-marker panels	Limited understanding of specific metabolite roles, standardization challenges
	Urine/Stool samples miRNAs	Non-invasive, early detection potential, Indicates the conditions of the intestines and the possibility of detecting cancer at an early stage.	Low sensitivity and specificity

Abbreviations: AFP: alpha-fetoprotein; DCP: des-gamma-carboxy prothrombin; miRNAs: microRNAs; lncRNAs: Long noncoding RNAs; ctDNA: circulating tumor DNA; EV: extracellular Vesicles; HCC: hepatocellular carcinoma.

**Table 2 diagnostics-14-02278-t002:** A summary of specific bacterial profiles associated with different liver pathologies.

Disease/Condition/Model (Reference)	Bacterial Genera (Increase/Decrease)	Key Findings
HCC Progression (Mouse Model) [104]	Dysbiotic microbiota, *Akkermansia muciniphila* ↓	TLR4 activation → Increase in mMDSCs → Suppression of T-cells → Weakens the body’s immune response → Allows HCC to progress: reversible with antibiotics or *Akkermansia muciniphila*.
MASLD-related Cirrhosis with HCC vs. Without HCC [101,103]	*Bacteroides* ↑, *Enterococcus* ↑, *Ruminococcaceae* ↑, *Bifidobacterium* ↓	Increased inflammation markers (fecal calprotectin, IL-8, IL-13, etc.). Microbiota triggered immunosuppressive response: ↑ regulatory T cells, ↓ CD8+ T cell activity.
Early HCC vs. Cirrhosis [105,106]	*Actinobacteria* ↑, *Gemmiger* ↑, *Parabacteroides* ↑, *Lipopolysaccharide-producing bacteria* ↑, *Butyrate-producing bacteria* ↓.	Increased microbial diversity, potential for non-invasive diagnostics. A model with 30 microbial markers had AUROC of 80.6%, distinguishing early HCC from non-HCC.
Cirrhotic Cases with HCC vs. Without HCC and Healthy Controls [102]	*Clostridium* ↑, *CF231* ↑, *Alphaproteobacteria ↓.*	Cirrhotic cases (with/without HCC) had lower bacterial richness than healthy individuals. Key classifiers of HCC–cirrhosis from healthy controls: Veillonella dispar, Faecalibacterium prausnitzii, Ruminococcus gnavus.
HCC vs. their healthy first-degree relatives [109]	*Lachnospiraceae* ↑, *Veillonella* ↑, *Ruminococcaceae UCG-014* ↑, *Peptostreptococcaceae* ↓, *Romboutsia* ↓, *Citrobacter* ↓.	Gut microbial composition in HCC patients is significantly altered. Romboutsia, Veillonella, and Peptostreptococcacae are potential biomarkers for HCC detection
Early vs. Middle vs. Advanced Liver cancer [110]	*Early: Clostridiales* ↑, *Firmicutes* ↑, *Streptococcus* ↑. *Middle: Ruminococcaceae* ↑, *Pasteurellaceae* ↑, *Tanticharoenia* ↑, *Vagococcus* ↑. *Advanced: Bifidobacteriales* ↑, *Actinobacteria* ↑, *Barnesiella* ↑, *Porphyromonadaceae* ↑, *Pseudomonadales* ↑.	Changes in microbiota with liver cancer progression: *Barnesiella* increased, *Ruminococcaceae* decreased.
HCC in elderly patients (60–80 years-old) [111]	*↓: A Blautia*, *Fusicatenibacter*, *Anaerostipes*, *CAG-56*, *Eggerthella*, *Lachnospiraceae_FCS020_group*, *Olsenella*. *↑: Escherichia-Shigella*, *Fusobacterium*, *Megasphaera*, *Veillonella*, *Tyzzerella_4*, *Prevotella_2*, *Cronobacter*	Age affects gut microbiota composition in HCC cases, and specific microbiota can be used as indicators for screening and diagnosing changes in elderly HCC patients.
Cirrhotic HCV Cases with HCC vs. Without HCC and Control [100]	*Bacteroides* ↑, *Lactobacilli* ↑, *Prevotella* ↓, *Prevotella/Bacteroides* ↓.	HCV-related cirrhosis and HCC show microbial dysbiosis, with HCC patients having higher proinflammatory bacteria compared to cirrhosis.
Viral HCC vs. Non-Viral HCC [112]	Viral HCC: *Faecalibacterium* ↑, *Agathobacter* ↑, *Coprococcus* ↑. Non-Viral HCC: *Bacteroides* ↑, *Streptococcus* ↑, *Ruminococcus gnavus* ↑, *Parabacteroides* ↑, *Erysipelatoclostridium* ↑. Short-chain fatty acid-producing bacteria ↓.	Gut dysbiosis linked to hepatocarcinogenesis and varies by HCC etiology. Microbiota signatures distinguish Viral-HCC and non-Viral HCC, offering potential for diagnosis and therapy.
HBV-related HCC vs. HBV-related Cirrhosis [113]	*Veillonella* ↓, *Streptococcus* ↓, *Fusobacterium* ↓, *Blautia* ↑, *Agathobacter* ↑	Certain bacterial genera may drive progression from cirrhosis to HCC in HBV cases, with gut microbiome showing potential for early HCC diagnosis.
HCC vs. iCCA [114]	iCCA: *Ruminococcus gnavus* ↓, *Veillonella* ↑. HCC: *Blautia* ↑.	Greater gut microbiome heterogeneity in iCCA vs. HCC and healthy controls. High *Veillonella* in iCCA linked to amino acid biosynthesis and glycolysis, while *Blautia* in HCC linked to phospholipid and thiamine metabolism.

Abbreviations: HCC: hepatocellular carcinoma; TLR4: Toll-like receptor 4; mMDSCs: monocytic myeloid-derived suppressor cells; MASLD: metabolic dysfunction-associated steatotic liver disease; AUROC: area under the receiver operating characteristic curve; HBV: hepatitis B virus; HCV: hepatitis C virus; iCCA: intrahepatic cholangiocarcinoma.↓: Decrease; ↑: Increase; →: Leads to or results in.

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
