# Peer review of "Novel Biomarkers for Early Detection of Hepatocellular Carcinoma"

_diagnostics, 2024, doi:10.3390/diagnostics14202278_

Round 1

Reviewer 1 Report

Comments and Suggestions for Authors

The paper is a comprehensive literature review that analyzes protein, circulating nucleic acid, metabolite, and quantitative proteomics-based biomarkers, evaluating the advantages and limitations of each approach in HCC. The topic is relevant because HCC is a frequent cancer and NAFLD and NASH are two relevant conditions.

The article is well organized and the english is fine. 

I suggest to insert a section like "materials and methods" where the authors could describe the criteria of articles' selection and the final number of articles evaluted.

The section about gut microbioma is really interesting. A table that summarizes the most relevant specific bacterial profiles for different liver pathologies could be usefull.  

Author Response

We sincerely thank Reviewer 1 for the positive feedback on our manuscript and for highlighting the relevance of our review on biomarkers for hepatocellular carcinoma (HCC).

  1. Materials and Methods Section: We appreciate the suggestion to include a "Materials and Methods" section. We have now added this section to the manuscript.
  2. Gut Microbiome Section: We are glad to hear that the section on gut microbiota was found interesting. In response to your suggestion, we have included a new table summarizing the most relevant bacterial profiles associated with different liver pathologies, which we believe adds value to the manuscript.

Reviewer 2 Report

Comments and Suggestions for Authors

In this manuscript ‘Novel Biomarkers for Early Detection of Hepatocellular Carcinoma’, it is a valuable review in this field.  Some concerns should be further addressed.

1.     Firstly,For example, in the lines 155-158: With the emergence of NGS and the advancements in precision medicine, omics data, such as genomic, epigenomic, transcriptomic, proteomic, and metabolomic data, can detect biological heterogeneity, facilitating the discovery of novel HCC biomarkers, as seen in Figure 1. 158. As a noun, genomic, epigenomic, transcriptomic, proteomic, and metabolomic commonly were written as genomics, epigenomics, transcriptomics, proteomics, and metabolomics.

2.     Secondly, in the lines 159, figure 1 title should be on the bottom of the figure, not on the top.

3.     In the lines 164-166: (2) Tumor Protein Markers: Proteins produced by tumor cells can serve as biomarkers detectable in blood samples. (3) EVs, RBCs, CTCs, and Lymphocytes: These circulating components can carry tumor-specific markers, which can be analyzed to detect the presence of HCC. Please provide the full name of the EV, RBC and CTC. They are not friendly to readers.

4.     Lastly, in the last column and last row of the Table 1, Low sensitivity, specificity may be Low sensitivity and specificity.

Urine/Stool samples miRNAs

Non-invasive, early detection po-tential, Indicates the conditions of the intestines and the possibility of detecting cancer at an early stage.

Low sensitivity, specificity

Comments on the Quality of English Language

In this manuscript ‘Novel Biomarkers for Early Detection of Hepatocellular Carcinoma’, it is a valuable review in this field.  Some concerns should be further addressed.

1.     Firstly,For example, in the lines 155-158: With the emergence of NGS and the advancements in precision medicine, omics data, such as genomic, epigenomic, transcriptomic, proteomic, and metabolomic data, can detect biological heterogeneity, facilitating the discovery of novel HCC biomarkers, as seen in Figure 1. 158. As a noun, genomic, epigenomic, transcriptomic, proteomic, and metabolomic commonly were written as genomics, epigenomics, transcriptomics, proteomics, and metabolomics.

2.     Secondly, in the lines 159, figure 1 title should be on the bottom of the figure, not on the top.

3.     In the lines 164-166: (2) Tumor Protein Markers: Proteins produced by tumor cells can serve as biomarkers detectable in blood samples. (3) EVs, RBCs, CTCs, and Lymphocytes: These circulating components can carry tumor-specific markers, which can be analyzed to detect the presence of HCC. Please provide the full name of the EV, RBC and CTC. They are not friendly to readers.

4.     Lastly, in the last column and last row of the Table 1, Low sensitivity, specificity may be Low sensitivity and specificity.

Urine/Stool samples miRNAs

Non-invasive, early detection po-tential, Indicates the conditions of the intestines and the possibility of detecting cancer at an early stage.

Low sensitivity, specificity

Author Response

We thank Reviewer 2 for the constructive feedback on our manuscript. Your comments have helped us improve the clarity and quality of our work.

  1. Terminology Adjustments: We have revised lines 155-158 to reflect the correct nomenclature, changing "genomic, epigenomic, transcriptomic, proteomic, and metabolomic" to "genomics, epigenomics, transcriptomics, proteomics, and metabolomics" to align with standard terminology.
  2. Figure Title Placement: The title of Figure 1 has been moved to the bottom of the figure as suggested.
  3. Full Terminology Inclusion: In lines 164-166, we have expanded the abbreviations for Extracellular Vesicles (EVs), Red Blood Cells (RBCs), and Circulating Tumor Cells (CTCs) to improve readability for our audience.
  4. Table Correction: We have corrected the phrase in the last column and last row of Table 1 from "Low sensitivity, specificity" to "Low sensitivity and specificity" for clarity.